

# Ship-based estimates of momentum transfer coefficient over sea ice and recommendations for its parameterization

Piyush Srivastava[1,5], Ian M. Brooks[1], John Prytherch[2,6], Dominic J. Salisbury[1], Andrew D. Elvidge[3], Ian A. Renfrew[3] and Margaret J. Yelland[4]

[1]School of Earth & Environment, University of Leeds, Leeds, LS2 9JT, UK

[2]Department of Meteorology, University of Stockholm, Stockholm, Sweden

[3]School of Environmental Sciences, University of East Anglia, Norwich, UK

[4] National Oceanography Centre, Southampton, UK

[5]Centre of Excellence in Disaster and Mitigation and Management, Indian Institute of Technology, Roorkee, India

[6] Bolin Centre for Climate Research, University of Stockholm, Stockholm, Sweden

Corresponding author: Piyush Srivastava (p.srivastava@leeds.ac.uk)



## Abstract

A major source of uncertainty in both climate projections and seasonal forecasting of sea ice is
inadequate representation of surface–atmosphere exchange processes. The observations needed to
improve understanding and reduce uncertainty in surface exchange parameterizations are challenging
to make and rare. Here we present a large dataset of ship-based measurements of surface momentum
exchange (surface drag) in the vicinity of sea ice from the Arctic Clouds in Summer Experiment
(ACSE) in July-October 2014, and the Arctic Ocean 2016 experiment (AO2016) in August–September
2016. The combined dataset provides an extensive record of momentum flux over a wide range of
surface conditions spanning the late summer melt and early autumn freeze-up periods, and a wide
range of atmospheric stabilities. Surface exchange coefficients are estimated from in situ eddy
covariance measurements. The local sea-ice fraction is determined via automated processing of
imagery from ship-mounted cameras. The surface drag coefficient, $C_{D10n}$, peaks at local ice fractions
of 0.6–0.8, consistent with both recent aircraft-based observations and theory. Two state-of-the-art
parameterizations have been tuned to our observations with both providing excellent fits to the
measurements.



## 1    Introduction

The Arctic region is changing rapidly. Surface temperatures are rising at a rate more than twice the planetary average, a process known as Arctic Amplification (Serreze and Barry, 2011; Cohen et al., 2014; Stuecker et al., 2018; Dai et al., 2019). Such rapid warming is drastically altering the physical landscape of the Arctic, most visibly the dramatic reduction in sea-ice extent (Onarheim et al., 2018), thickness, and age (Ricker et al., 2017; Kwok, 2018), and has the potential to impact a host of biological and chemical processes (Howes et al., 2015; Lehnherr et al., 2018). Changes in the Arctic may also impact lower latitudes via modification of weather patterns and ocean circulation (Cohen et al., 2014; Overland et al., 2016).

Whilst climate models robustly reproduce Arctic amplification, they have been less successful in making accurate seasonal forecasts of sea-ice extent (Stroeve et al., 2014) or even capturing the observed sea ice decline over the past decades (Stroeve et al., 2012). There is also large inter-model variability in projections of future climate over varying timescales (Hodson et al., 2013; Stroeve et al., 2014; Zampieri et al., 2018). A major source of uncertainty in models is the representation of turbulence-driven surface exchanges (Bourassa et al., 2013; Vihma et al., 2014; Tsamados et al., 2014; LeMone et al., 2018). Turbulent exchange is a subgrid-scale process parameterized in terms of resolved model variables and surface transfer coefficients. A lack of observational data in high-latitude environments has resulted in large uncertainty in the parameterization of the transfer coefficients of momentum ($C_D$), heat ($C_H$) and moisture ($C_E$). Here, we focus on the parameterization of the momentum transfer (drag) coefficient, $C_D$.

The exchange of momentum between the atmosphere and sea ice directly affects the dynamical evolution of both the atmospheric boundary layer and the sea ice. The exchange is partly dependent on physical properties of the surface. With ongoing sea ice loss and the increasing spatial extent of the Arctic Ocean's marginal ice zone (MIZ) (Strong and Rigor, 2013; Rolph et al., 2020), the nature of this exchange is subject to change, implying improved understanding of the physical processes is critical. Recent studies have shown that the model reproduction of future sea-ice thickness and extent (Rae et al., 2014; Tsamados et al., 2014), the near-surface atmosphere (Rae et al., 2014; Renfrew et al., 2019), and the polar ocean (Stoessel et al., 2008; Roy et al., 2015) are all sensitive to the parameterization of surface momentum exchange over sea ice.

Most models have rather simplified approaches for parameterizing the transfer coefficients over sea ice: prescribing either a constant value for equivalent neutral transfer coefficients for all sea ice, or two





different values, corresponding to the MIZ and pack ice conditions along with empirical ice
morphological parameters (Notz, 2012; Lüpkes et al., 2013; Elvidge et al., 2016). They then typically
utilize a classical 'mosaic' or 'flux averaging' approach where fluxes are estimated separately over sea
ice and open water for each grid box and an 'effective' turbulent flux is calculated as the weighted
average using the fractions of open water and sea ice (Claussen, 1990; Vihma, 1995).
For models that assume a fixed $C_{D10n}$ over ice, the flux averaging method leads to a monotonically
increasing $C_{D10n}$ across the MIZ; this is not supported by observations (Mai et al., 1996; Hartman et
al., 1994; Schroder et al., 2003; Andreas et al., 2010; Elvidge et al., 2016) which indicate a peak at ice
fractions of 50-80%. The value of sea-ice concentration at which $C_{D10n}$ peaks depends upon the ice-
morphology (Elvidge et al., 2016). It arises due to the contribution of form drag at the edges of floes,
leads, melt ponds and ridges (Arya 1973, 1975; Andreas et al., 2010; Lüpkes et al., 2012, Lüpkes and
Gryanik, 2015; Elvidge et al., 2016).
Andreas et al. (2010) suggested a simple empirically-based parameterization of $C_{D10n}$ in terms of a
quadratic function of ice concentration. Based on theoretical considerations (Arya, 1973; Arya, 1975;
Hanssen-Bauer and Gjessing, 1988; Garbrecht et al., 2002; Birnbaum and Lüpkes, 2002; Lüpkes and
Birnbaum, 2005), Lüpkes et al. (2012; L2012 hereafter) developed a physically-based hierarchical
parameterization for $C_{D10n}$ which, in its lowest level of complexity, requires only ice fraction as the
independent variable. The L2012 parameterization scheme qualitatively reproduces the observed peak
in $C_{D10n}$ over the MIZ.
Recently, Elvidge et al. (2016; E2016) used aircraft measurements over the Arctic MIZ to develop a
data set of 195 independent estimates of $C_{D10n}$ over the MIZ, more than doubling the number of
observations previously available. Their observations were consistent with the theory of L2012;
however, they found a large variation in $C_{D10ni}$ ($C_{D10n}$ for 100% ice cover) demonstrating that this
depends strongly on ice morphology – as also found by Castellani et al. (2014) who applied bulk
parameterizations to ice morphology data based on laser altimetry.  E2016 recommended modified
values of key parameters in the L2012 scheme and, subsequently, this scheme with these settings has
been implemented in the Met Office Unified Model (MetUM). Renfrew et al. (2019) demonstrated
that this new scheme significantly reduced biases and root-mean-square errors in the simulated wind
speed, air temperature and momentum flux over, and downstream of, the MIZ; in addition to having
widespread impacts throughout the Arctic and Antarctic via, e.g., mean sea level pressure. The new
scheme became part of the operational forecasting system at the Met Office in September 2018 and





part of the latest climate model configuration (in GL8). However, at present a constant value of $C_{\mathrm{D10ni}}$
is used; a known limitation in the veracity of surface momentum exchange over sea ice.
At present, the complexity of physically-based parameterizations of momentum exchange over sea ice
exceeds the parameterization constraints provided by observations. In other words, despite recent
progress, we are still lacking the observational data sets required for further parameterization
development. Here, we utilize a large data set of ship-based measurements of surface momentum
exchange – made as part of the Arctic Clouds in Summer Experiment (ACSE) in July-October 2014,
and the Arctic Ocean 2016 expedition (AO2016) in August-September 2016 – to study momentum
exchange over heterogeneous sea ice.  We investigate the relationship between surface drag and sea-
ice concentration within the existing framework suggested by L2012, E2016, and its recent extension
by Lüpkes and Gryanik (2015; L2015) using over 500 new estimates of surface drag and local sea-ice
concentration measurements derived from on-board imagery, over varying sea-ice conditions and a
range of near-surface atmospheric stabilities.

## 2      Parameterization background
The surface flux of momentum is $\tau = -\rho u_*^2 = \rho C_D U^2$, where $\rho$ is the air density, $u_*$ is the friction
velocity, and $U$ is the wind speed at a reference height. The drag coefficient, $C_{\mathrm{D}}$, is derived from
Monin-Obukhov similarity theory (MOST, Monin and Obukhov, 1954) as:

$$C_D = \kappa^2 [ln(z/z_0) - \psi_m(\zeta)]^{-2} \qquad (1)$$

Here, $\kappa$ is the von Kármán constant, $z$ is the reference height at which the transfer coefficient is
evaluated, $z_0$ is the aerodynamic roughness length, and $\psi_m$ is an integrated stability correction function
(Stull, 1990).
Over land surfaces, the aerodynamic roughness length is in general taken as constant depending upon
the surface characteristics, while over the open water the roughness length varies with wind speed and
is typically parameterized using a type of Charnock relation (Charnock, 1955). When the surface
consists of a mix of ice and open water, an effective turbulent flux over the area is usually calculated
by taking a weighted average over the fraction of open water and sea ice (Vihma, 1995):

$$C_{D10n} = (1 - A)C_{D10nw} + AC_{D10ni}. \qquad (2)$$



Here, $C_{D10nw}$ and $C_{D10ni}$ are, respectively, the neutral transfer coefficients for momentum over water
and ice surfaces, and $A$ is the fraction of the surface covered by ice. Over sea ice, an additional drag
contribution, the form drag, $C_{D10nf}$, is generated due to air-flow pressure against the edges of floes,
leads, and melt ponds (Andreas et al., 2010; L2012; L2015; E2016). The overall equivalent neutral
drag coefficient is then given by

$$C_{D10n} = (1-A)C_{D10nw} + AC_{D10ni} + C_{D10nf}. \qquad (3)$$

Lüpkes et al. (2012) proposed a hierarchical parameterization for $C_{D10n}$ in which form drag, in its
lowest level of complexity, is parameterized as a function of ice fraction only

$$C_{D10nf} = A \frac{h_f}{D_i} S_c^2 \frac{c_e}{2} \left[ \frac{ln^2(h_f/z_{0w})}{ln^2(10/z_{0w})} \right]. \qquad (4)$$

Here, $D_i$ is the characteristic length scale of the floe, $h_f$ is the freeboard height, $S_c$ is the sheltering
function, and $c_e$ is the effective resistance coefficient. Lüpkes et al. (2012) provided simplified forms
for these parameters either in terms of ice fraction or as constants:

$$S_c = \left(1 - exp\left(-s\frac{D_w}{h_f}\right)\right), \qquad (5)$$

where

$$D_w = D_i \left(1 - \sqrt{A}\right)/\sqrt{A}, \qquad (6)$$

$$h_f = h_{max}A + h_{min}(1-A), \qquad (7)$$

$$D_i = D_{min} \left(\frac{A_*}{A_*-A}\right)^{\beta}, \qquad (8)$$

and

$$A_* = \frac{1}{1-(D_{min}/D_{max})^{1/\beta}}. \qquad (9)$$

For operational purposes, L2012 suggested optimum values of the parameters used in the above
expressions (Table 1). E2016 evaluated the L2012 scheme with in situ aircraft measurements and
found that with slightly modified values of these key parameters (Table 1), it represented the behaviour
of $C_{D10n}$ well.
The L2012 scheme assumes that the wind profile is always adjusted to the local surface. However, this
assumption is not necessarily valid where the surface conditions change over small spatial scales, and



|  | $c_e$ | $S_c$ | $h_f$ | $\beta$ | $h_{max}$ | $h_{min}$ |
|---|---|---|---|---|---|---|
| **L2012** | 0.30 | Eq. (5) with $s = 0.5$ | Eq. (7) | 1 | 0.534 m | 0.286 m |
| **E2016A** | 0.17 | Eq. (5) with $s = 0.5$ | Eq. (7) | 1 | 0.534 m | 0.286 m |
| **E2016B** | 0.10 | Eq. (5) with $s = 0.5$ | Eq. (7) | 0.2 | 0.534 m | 0.286 m |
| **ACSE+AO2016 (P2021-L2012)** | 0.10 | Eq. (5) with $s = 0.5$ | Eq. (7) | 1 | 0.534 m | 0.286 m |
|  |  |  |  |  |  |  |
| **L2015** | 0.4 | 1 | 0.41 m | 1.4 | - | - |
| **L2015 (ACSE+AO2016) (P2021-L2015)** | 0.18 | 1 | 0.41 m | 1.1 | - | - |
|  |  |  |  |  |  |  |
| **L2015 (ACSE data only)** | 0.22 | 1 | 0.41 m | 1.0 | - | - |
| **L2015 (AO2016 data only)** | 0.18 | 1 | 0.41 m | 1.1 | - | - |


**Table 1:** Parameter settings for the form drag component of the L2012 scheme (Lüpkes et al., 2012, rows 1–4): as recommended in L2012, E2016A and E2016B, P2021-L2012; and the L2015 scheme (Lüpkes and Gryanik, 2015, rows 5–8): as recommended in L2015, and fit to ACSE+AO2016, ACSE only, AO2016 only. The L2012 variants use: $D_{min} = 8$ m and $D_{max} = 300$ m, while the L2015 variants use $D_{min} = 300$ m. The primary tuning parameter is the effective resistance coefficient, $c_e$, while $\beta$ has a second order effect on the shape of the curve. $S_c$ and $h_f$ were tunable parameters in L2012, but found by L2015 to have marginal impact and set as constants for simplicity.

161

the fetch over the local surface is insufficient for the wind profile to come into equilibrium with its characteristics. To overcome this issue in the existing schemes, L2015 suggested a fetch-dependent parameterization of the form drag component of the total drag at arbitrary height:

$$C_{Dnf} = C_{Dnf,w}(1 - A) + C_{Dnf,i}A, \qquad (10)$$

where $C_{Dnf,w}$ and $C_{Dnf,i}$ are, respectively, the neutral form-drag coefficients related to the fetch over open water and over ice, and are expressed as:

$$C_{Dnf,k} = A \frac{h_f}{D_i} S_{c,k}^2 \frac{c_e}{2} \left[ \frac{ln^2(h_f/ez_{0,k})}{ln^2(z_p/z_{0,k})} \right], where\ k = i, w \qquad (11)$$



Thus, the L2015 scheme incorporates two form drag contributions and both are weighted by their
respective surface fractions. Equation (11) differs from the formulation in L2012 (equation 4) only by
the inclusion of the Eulerian number, $e$, in the logarithmic term of the numerator, consistent with
previous work by other groups (e.g., Hanssen-Bauer and Gjessing, 1988), and is valid for any reference
height, $z_p$. Here we evaluate L2012 and L2015 against in situ estimates of $C_{D10n}$ to assess the impact
of including form drag for both water and ice surfaces. We do not evaluate the higher levels of
complexity in L2015. The values of various parameters used in the L2015 parameterization, both as
originally published and tuned to our observations, are given in Table 1, along with those for L2012
and E2016.

## 3     Measurement and methods

### 3.1     Field Measurements

We utilize data from two field campaigns, the Arctic Cloud in Summer Experiment (ACSE, Tjernström
et al., 2015, 2019; Achtert et al., 2020), part of the Swedish-Russian-US Arctic Ocean Investigation
on Climate-Cryosphere-Carbon (SWERUS-C3), and the Arctic-Ocean 2016 (AO2016) expedition.
Both ACSE and AO2016 took place on board the Swedish icebreaker *Oden*. The ACSE cruise took
place between 5 July and 5 October 2014, starting and ending in Tromsø, Norway, and working around
the Siberian shelf, through the Kara, Laptev, East Siberian and Chukchi seas (Fig. 1). A change of
crew and science team took place in Utqiaġvik (formerly Barrow), Alaska on 20 August. The AO2016
expedition was carried out between 8 August and 19 September 2016 in the central Arctic Ocean,
starting from, and returning to, Longyearbyen, Svalbard (Fig. 1).

### 3.2     Surface turbulence and meteorological measurements

Turbulent fluxes were measured with an eddy covariance system installed at the top of *Oden's*
foremast, 20.3 m above the waterline. On ACSE this consisted of a Metek USA-100 sonic anemometer
with heated sensing heads, a Li-COR Li-7500 open path gas analyser, and an Xsens MTi-700-G motion
sensing package installed at the base of the anemometer. The ship's absolute heading and velocity
were obtained from its navigation system. The Metek sonic anemometer failed at the start of AO2016
and was replaced with a Gill R3 sonic anemometer. The raw turbulent wind components, at 20 Hz,
were corrected for platform motion following Edson et al. (1998) and Prytherch et al. (2015).
Corrections for flow distortion of the mean wind were derived from a CFD model (Moat et al. 2015).
Turbulent fluxes of heat, momentum and moisture were estimated by the eddy-covariance technique
over 30-min averaging intervals.



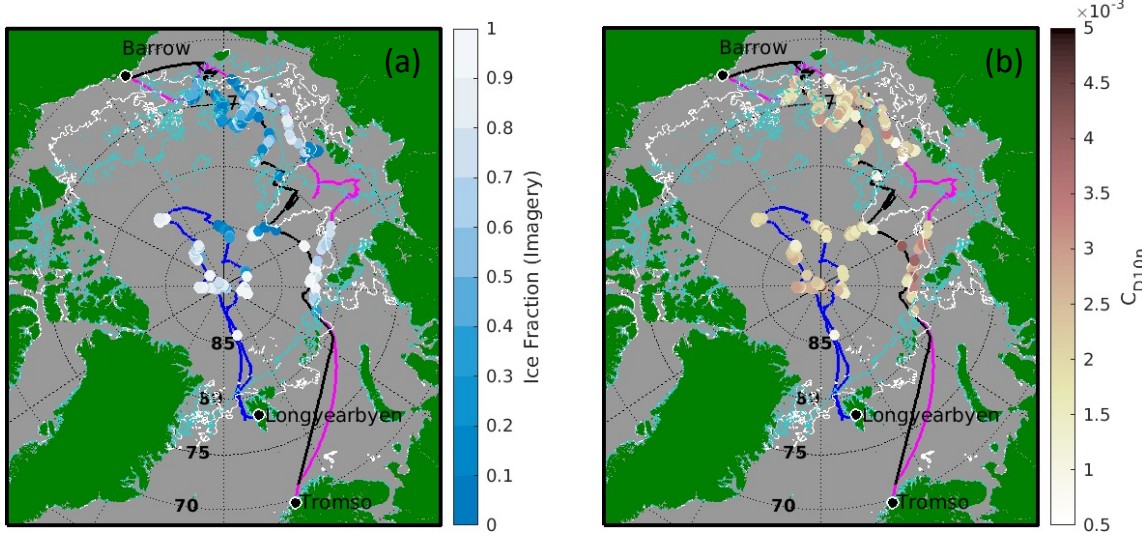



**Figure 1.** The cruise tracks of ACSE (leg 1, 5 July – 18 August 2014 (magenta) and leg 2, 21 August – 5 October
(black) and AO2016 (8 August – 19 September 2016, (blue)) with (a) sea-ice fraction from in situ imagery and
(b) $C_{D10n}$, for each 30-min flux period shown. The sea-ice extent from AMSR2 on 7 August 2014 (white) and
2016 (cyan) – about midway through ACSE and at the start of AO2016 – are shown for reference and to give
an indication of the variability between years.


Mean temperature ($T$) and relative humidity ($RH$) at the mast top were measured with an aspirated
sensor – a Rotronic T/RH sensor during ACSE and a Vaisala HMP-110 during AO2016. Additional
$T$, $RH$ and pressure ($P$) measurements were made by a Vaisala PTU300 sensor on the 7th deck of the
ship. Pressure at the mast top was obtained by height-adjusting the measurement from the 7th deck.
The surface skin temperature was obtained from two Heitronics KT15 infra-red surface temperature
sensors, mounted above the bridge and viewing the surface on either side of the ship. Digital imagery
of the surface around the ship was obtained from 2 Mobotix M24 IP-cameras mounted on the port and
starboard beam rails above the bridge, approximately 25 m above the surface. Images were recorded
at 1-minute intervals during ACSE and 15-second intervals during AO2016. Profiles of atmospheric
thermodynamic structure and winds were obtained from Vaisala RS92 radiosondes, launched every 6
hours throughout both cruises. During ACSE a Radiometer Physics HATPRO scanning microwave
radiometer provided additional retrievals of lower-atmosphere temperature profiles every 5 minutes.
**3.3     Estimation of turbulence parameters and data screening**
The transfer coefficient of momentum is computed as:



$$C_{D10n} = \left(\frac{u_*}{U_{10n}}\right)^2 \qquad (12)$$

where $u_*$ is the measured friction velocity, and $U_{10n}$ is the 10 m equivalent neutral wind speed,
determined using Monin-Obukhov similarity theory and Businger-Dyer stability corrections (Businger
et al., 1971).
A total of 3421 and 1555 individual half hourly flux estimates were obtained during ACSE and
AO2016 respectively. Data were removed from the analysis if they failed a set of flux quality control
criteria (Foken and Wichura, 1996) resulting in a subset of 1804 (247) flux estimates. Additional
quality control criteria were applied to filter out data unreliable for analysis of transfer coefficients:
•  The relative wind direction was restricted to $\pm 120°$ from bow-on, where the flow is clear of
the ship's superstructure.
•  Data points where the stability parameter, $z/L$ (where $L$ is the Obukhov length) was greater
than 1 or less than –2 were removed to avoid the effects of strong stability and instability.
•  Sign constancy between the turbulent heat flux and mean gradients was enforced. The
Richardson number, $Ri_B$, and $z/L$ should always have same sign, while the sensible heat flux,
$H$, should be of opposite sign to $z/L$. Inconsistencies may arise due to combined measurement
uncertainties where the temperature gradient or $H$ are small.
•  Data were also removed where the 10-m wind speed was less than 3 m s⁻¹.
After initial quality control we have a total of 1403 and 162 half hourly flux estimates. For ACSE data
an additional quality control criterion was applied based on the boundary-layer profiles. Working
around the MIZ, ACSE experienced multiple warm air advection events. These result in strong near-
surface air mass modification and the formation of very low or surface-based temperature inversions
(Tjernström et al., 2019) which can exhibit significant spatial and temporal variability. Temperature
profiles from the HATPRO microwave radiometer, bias-corrected through extensive comparisons with
6-hourly radiosondes, were used to detect surface inversions with 5-minute temporal resolution
following Tjernström et al. (2019). The profiles were classified as surface-based inversions, low-level
inversions (inversion base height < 200 m), and well-mixed boundary layers (inversion base height >
200 m). The flux periods with mixed surface-based and low-level inversions were discarded from the
analysis on the basis that the change in near-surface thermodynamic structure was likely to
compromise the quality of the flux-profile relationship upon which the calculation of $C_{D10n}$ depends.
Following this step, we were left with 1051 data points from the ACSE campaign. No high-frequency
profile measurements were available from AO2016; however, operating much further from the ice



edge, AO2016 was not subject to the frequent warm air advection and air mass modification events
seen during ACSE.

## 3.4    Determination of sea-ice concentration
Estimates of sea-ice fraction are drawn from two sources: (i) a local estimate of ice fraction determined
from digital imagery from the ship; and (ii) daily ice fractions, derived from satellite based Advanced
Microwave Scanning Radiometer (AMSR2) passive microwave measurements (Spreen et al., 2008).
Our local raw imagery consists of high-definition (2048×1536) images of the surface to port and
starboard, obtained from Mobotix MX-M24M IP cameras mounted above the ship's bridge, 25 m
above the surface. Additional images from a camera pointed over the bow are used for visual inspection
while selecting the periods when *Oden* was in the ice, but not processed because the ship dominates
the near field of the image.
On-board imagery can provide local surface properties including sea ice and melt pond fractions with
a spatial resolution of order metres on a time base matched to the flux averaging time (Weissling et
al., 2009). The large volume of imagery sampled requires automated image processing techniques to
estimate ice properties (Miao et al., 2015; Perovich et al., 2002; Renner et al., 2014; Webster et al.,
2015; Wright and Polashenski, 2018). Here we use the Open Source Sea-ice Processing (OSSP)
algorithm of Wright and Polashenski (2018). The surface properties obtained during each 30-minute
interval are averaged to give the sea ice and melt pond fractions. The image processing methodology
is described in Appendix A. Limitations on quality and availability of imagery resulted in a further
206 (36) flux estimates for the ACSE (AO2016) datasets being discarded. After all quality control
criterion are applied and flux estimates matched with robust estimates of the local ice fraction, we
retain at total of 542 flux estimates: 416 from ACSE and 126 from AO2016. Initially melt ponds are
treated as open water; the impact of this is examined in Appendix B.
Satellite-based sea ice products are widely used to prescribe ice concentration in operational forecast
models and have been used to assess the dependence of in situ flux measurements as a function of ice
fraction (e.g., Prytherch et al., 2017). However, they have significant uncertainties when related to in
situ flux measurements due to their relatively coarse temporal and spatial resolution (Weissling et al.,
2009) resulting in a mismatch between the satellite footprint and that of the surface flux measurement,
and the times of the measurements. The AMSR2 satellite measurements used here provide *daily* sea-
ice concentration on a 6.25 km grid, while eddy covariance flux estimates are for 30-minute periods





and have a footprint of the order of a few hundred metres to a kilometre. The AMSR2 estimates are
interpolated spatially to the locations of each flux measurement.

## 4    Results
### 4.1    Atmospheric conditions during ACSE and AO2016
Figures 2 and 3 show the meteorological and surface conditions during ACSE and AO2016. The first
half of ACSE was dominated by relatively low winds, and surface temperatures close to 0°C when in
the ice; much warmer temperatures are associated with open coastal waters. The second half of the
cruise experienced higher, and more variable winds, associated with multiple low-pressure systems.
Temperatures first fell to the freezing point of salt water on day of year (DoY) 218, although
Sotiropoulou et al. (2016) identified the start of freeze up as DoY 241. AO2016 saw a shift from
relatively low surface air pressure, and mostly low winds to higher pressure and more variable winds
with frequent occurrence of high winds around DoY 237.
Out of the total of 542 flux estimates, we have 184, 282, and 76 flux estimates in stable ($z/L > 0.01$),
near-neutral ($-0.01 < z/L < 0.01$), and unstable conditions ($z/L < -0.01$) respectively. This distribution
in static stability augments the limited data sets already available over the marginal ice zone, which
have been predominantly in unstable conditions (e.g., E2016).
### 4.2    Ice surface characteristics
Figure 4 shows the variation over time of the ice, melt pond, and open water fractions determined from
the on-board imagery and from AMSR2. Note that the variations represent geographic variability along
the cruise tracks, as well as temporal changes in ice conditions. During the early phase of the ACSE
campaign, to mid-July (DoY 185 to 196), the ice encountered was mostly old ice (Tjernström et al.,
2019) with an average ice concentration of about 70%; melt ponds and open water had 11% and 18%
coverage respectively. In this phase the average concentration from AMSR2 was 95% – larger than
the sum of average local ice and melt pond concentrations (81%). From late July to early August (DoY
209 to 229), the average local ice concentration was 56%, with melt pond and open water fractions of
about 17% and 27%. Here the AMSR2 concentration was lower than that from imagery, at 44%.
During this period, warm continental air from Siberia flowed northward across the *Oden's* track
causing a rapid melting of ice (Tjernström et al., 2015; Tjernström et al., 2019). From late August to
mid-September (DoY 239 to 255), the average local concentration declined to 27.6% and the surface

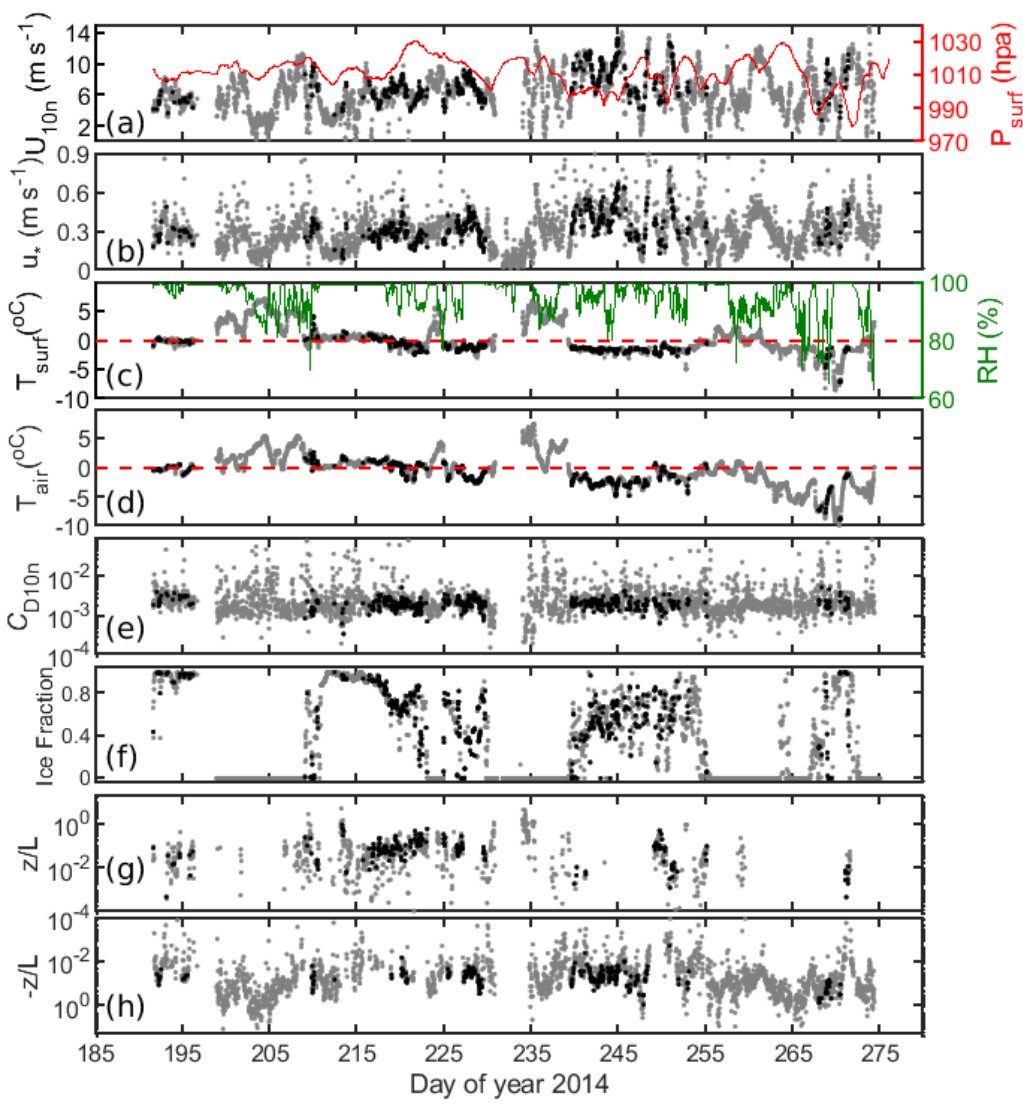



**Figure 2.** Time series of **(a)** 10 m neutral wind speed, $U_{10n}$ and Surface pressure, $P_{surf}$, (Secondary axis), **(b)** friction velocity, $u_*$ **(c)** surface temperature, $T_{surf}$ and relative humidity, RH,(Secondary axis), **(d)** air temperature, $T_{air}$, **(e)** 10 m equivalent neutral drag coefficient, $C_{D10n}$ **(f)** Ice fraction from AMSR2 satellite, **(g)** Monin-Obukhov stability parameter, $z/L$, for $z/L > 0$ (stable) and **(h)** $z/L < 0$ (unstable), for ACSE data. The grey dots are 30-minute flux periods from the whole cruise, while the black dots correspond to the flux data points that pass quality control. In panels **(c)** and **(d)**, the dashed red lines show $T_{air} = T_{surf} = 0$ºC.

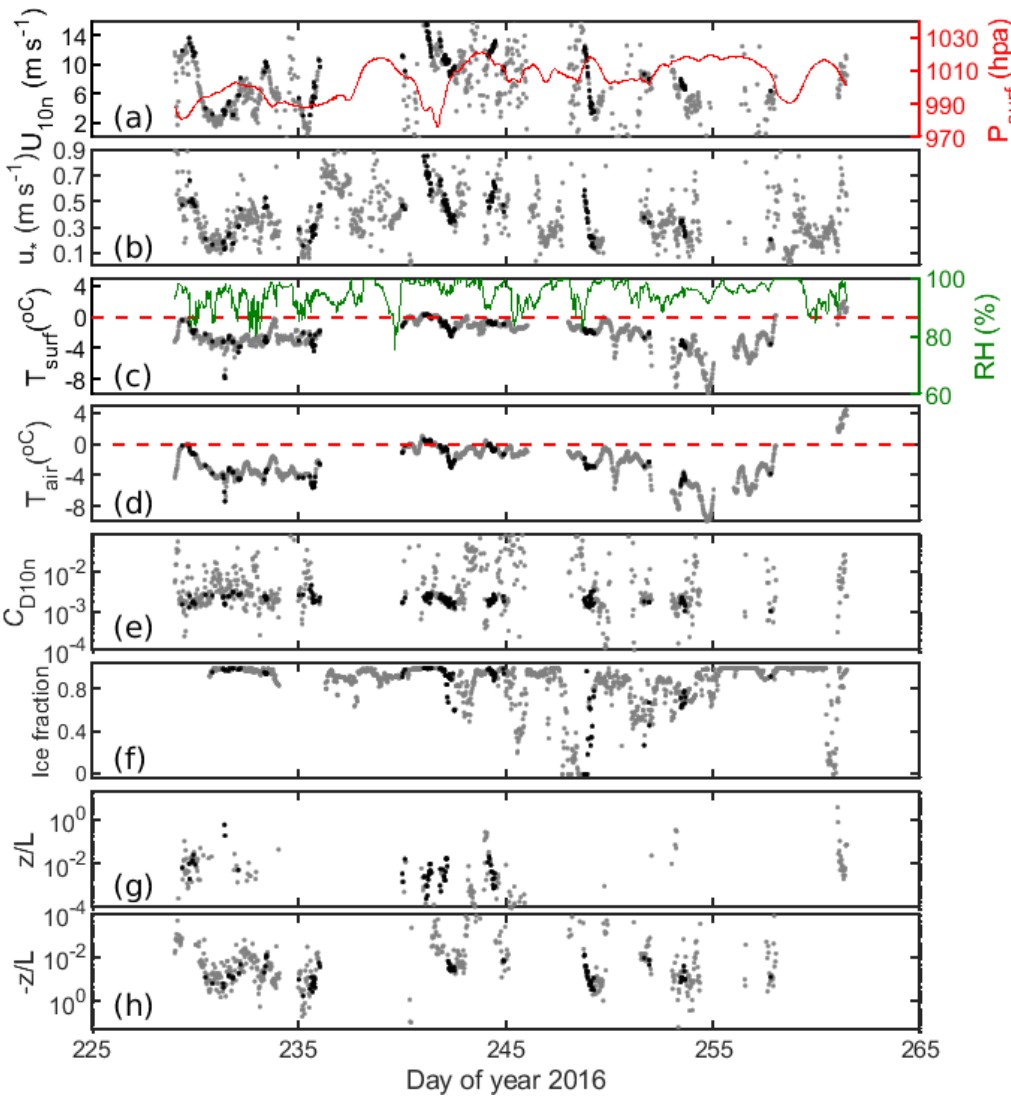

**Figure 3.** As Figure 2 but for AO2016.

was mostly characterized by large areas of open water (63%). The AMSR2 ice fraction was 54%, approximately twice that from the imagery. During late September (DoY 268 to 272), a large variation in the surface conditions was observed and often the ice concentration was higher than 90% due to the presence of newly formed thin ice, nilas, and pancake ice. During the AO2016 campaign, the surface was mostly characterized by old and thick ice with intermittent patches of thin ice and melt ponds,





reflecting the more northerly cruise location. The average ice concentrations from imagery and
AMSR2 were found to be about 80% and 90% respectively.

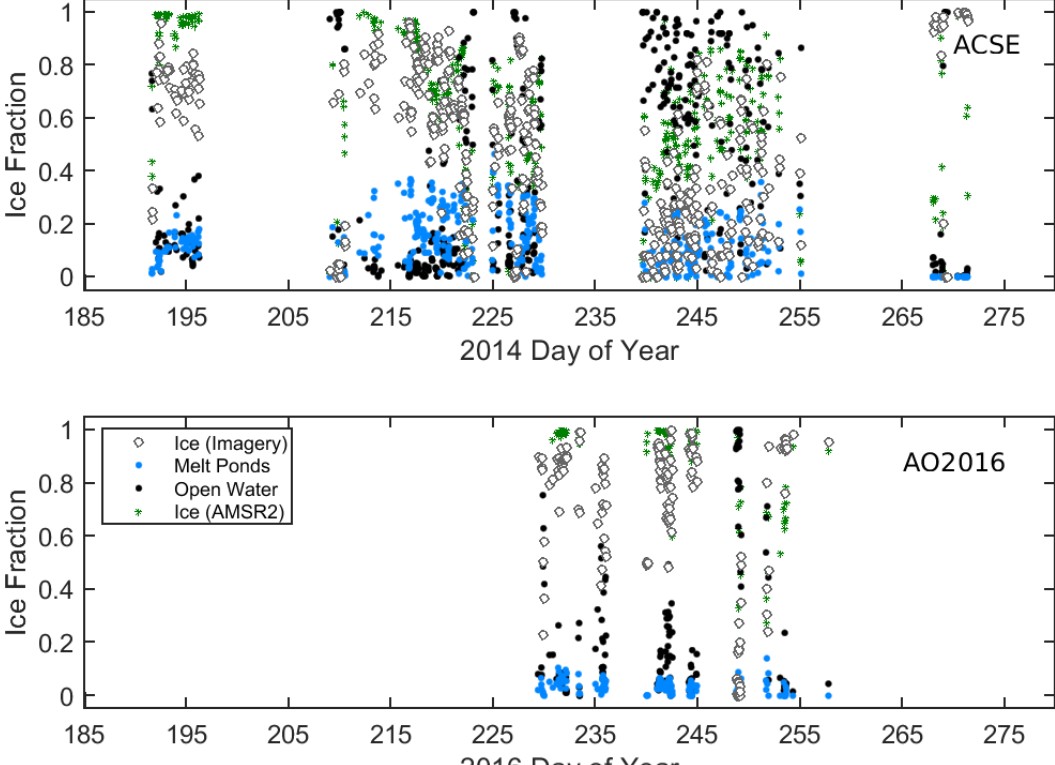

**Figure 4.** Time series of ice, melt pond and open water fractions (white, blue and black symbol respectively)
from the local imagery, and ice fraction (green) from AMSR2, interpolated to the ship location. The top panel
is for ACSE and the bottom panel for AO2016.

Figure 5 shows a direct comparison of ice fraction from the in situ imagery and AMSR2. There is a
broad correspondence, but a very high degree of scatter, and AMSR2 tends to overestimate the local
sea-ice fraction; the correlation coefficient, mean absolute bias, and root mean square error are 0.64,
0.21, and 0.28 respectively. It is clear from the ice concentration time series, however, that the bias
between AMSR2 and local ice fraction varies over time and appears to be related to the surface
conditions of melt or freeze up, in particular when changes are rapid. The largest difference between
ice fraction from both projects was found during the early freeze-up season where there is extensive
very thin ice.

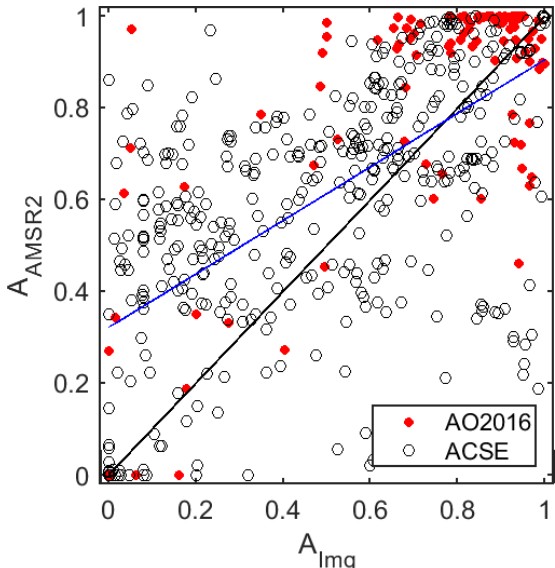

**Figure 5.** A comparison of ice fraction derived from the local imagery and from AMSR2 for both field campaigns. The linear regression ($A_{\text{AMSR}} = 0.584A_{\text{Img}}+0.321$) and 1:1 lines are shown in blue and black respectively.

## 4.3 Variation of momentum transfer coefficient with sea-ice concentration

We first assess the variability of surface drag with sea-ice fraction using local ice concentration from the on-board imagery (Fig. 6). The range and median values of $C_{\text{D10n}}$ over sea ice ($A_{\text{Img}} > 0$) are similar to those of previous studies (Banke and Smith, 1971; Overland et al, 1985; Guest and Davidson, 1987; Castellani et al., 2014; Elvidge et al., 2016). The peak $C_{\text{D10n}}$ is found at 0.6–0.8 ice fraction bin, consistent with Lüpkes et al. (2012) and Elvidge et al. (2016). The median values of $C_{\text{D10n}}$ in both datasets agree well for high ice fractions (Figs. 6b and 6c), however, there is insufficient AO2016 data for $A_{\text{Img}} < 0.5$ to make a robust comparison with ACSE. Given the good general agreement between ACSE and AO2016 we will only consider the joint data set from here on.

The measurements are compared with the L2012, L2015, and E2016 parameterization schemes. Note that these all require specified values of $C_{\text{D10n}}$ over open water ($A_{\text{Img}} = 0$) and solid ice ($A_{\text{Img}} = 1$) (see Eq. 2); these vary with conditions, dramatically so for $A_{\text{Img}} = 1$, as demonstrated by Elvidge et al. (2016). Here, we follow E2016 and fix the values of $C_{\text{D10nw}}$ and $C_{\text{D10ni}}$ used in the parameterizations to the measurements, using the observed median values at $A_{\text{Img}} = 0$ and $A_{\text{Img}} > 0.8$ respectively for each

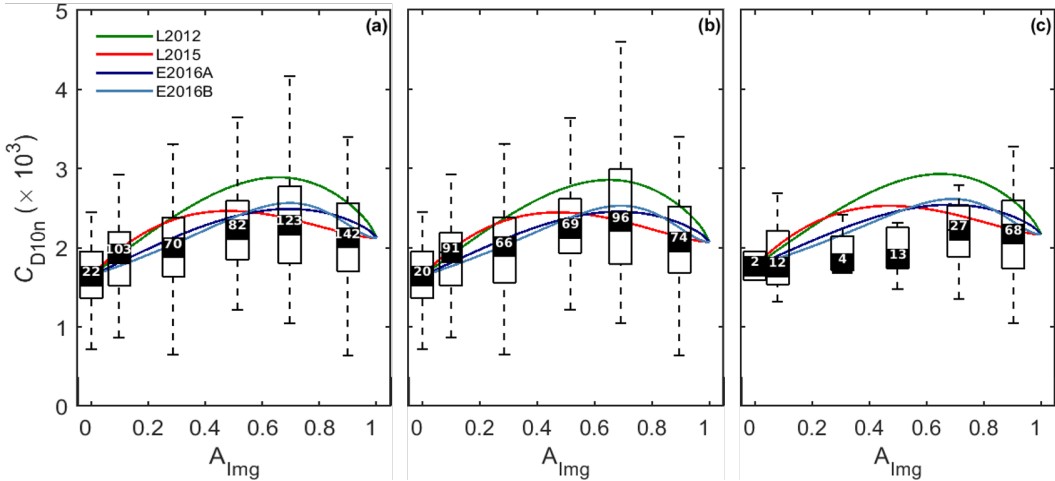

366

**Figure 6.** $C_{D10n}$ as a function of ice fraction as derived from local imagery ($A_{Img}$) for (a) the joint ACSE and
AO2016 datasets (n = 542), (b) the ACSE dataset (n = 416), (c) the AO2016 dataset (n = 126). The boxes show
the interquartile range and the bin median (black square) for bins of width = 0.2 and plotted at the mean ice
fraction for the bin; the number of data points in each bin is noted at the median level. Whiskers indicate the
range of the estimates, excluding any outliers, which are plotted individually if present. Parameterization
schemes are overlain as indicated, with each curve anchored at the observed median values of $C_{D10nw}$ ($A_{Img} = 0$)
and $C_{D10ni}$ (defined here as $A_{Img} > 0.8$) for each data set.


data set. Note that $A > 0.8$ is used, as opposed to $A = 1$, as navigational consideration meant the ship
rarely operated in regions with an ice fraction of 1. The mean ice fraction in this bin is 0.89. Here, we
do not adjust any of the other tuneable parameters in these parameterizations.
L2012 overestimates the observations of all but the lowest ice concentrations. E2016a and E2016b –
which follow L2012 with settings tuned to measurements over the MIZ from Fram Strait and the
Barents Sea – correspond well with the observations, with only a slight overestimate of the peak values.
L2015, which accounts for form drag over water as well as over ice, is a close match to the observations
for $A_{Img} > 0.6$ but overestimates the 0.2–0.4 and 0.4–0.6 bins and peaks at too low an ice concentration.
Note we will tune the L2015 scheme using our measurements in section 4.4.
The median value of $C_{D10n}$ at $A = 0$ was $1.65 \times 10^{-3}$, which is higher than those typically found over the
open ocean (Smith, 1980; Large and Yeager, 2009; Andreas et al., 2012). This may be a result of the
open water measurements being made under fetch-limited conditions close to the ice edge, or within
regions of open water within the pack ice, where an under-developed wave state may result in higher
drag (Drennan et al., 2003). We cannot, however, exclude the possibility that they result from an





incomplete correction for flow distortion over the ship (Yelland et al. 1998, 2002), or that the flux
footprint includes flow over nearby ice that is not visible in the imagery.
Figure 7 shows $C_{D10n}$ as a function of AMSR2 ice fraction. There is broad agreement with the values
in Fig 6a at low and high ice concentrations, but there is no peak in $C_{D10n}$ at intermediate
concentrations. Instead, the measurements with higher drags have moved to either lower or higher ice
fraction bins. This is not consistent with either our in situ imagery or previous aircraft-based studies,
suggesting it is a limitation of the AMSR2 imagery.

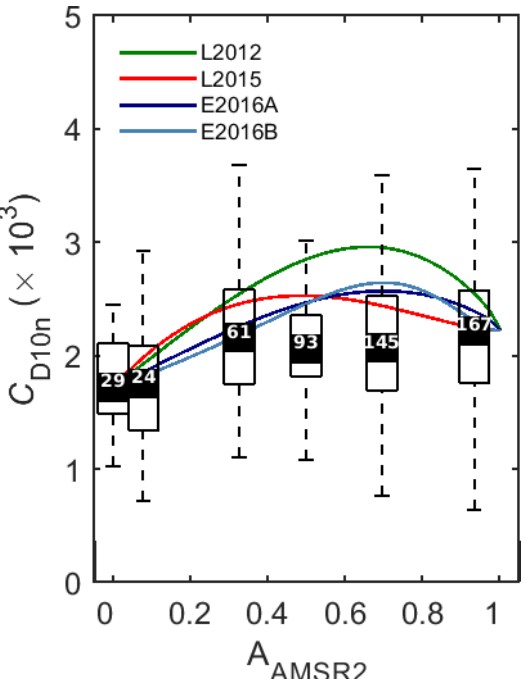


**Figure 7.** As Figure 6a but for ice fraction derived from AMSR2 satellite.

## 4.4    Updating parameterizations using local sea-ice concentration measurements
L2015 extended the parameterization of L2012 to explicitly represent the impact of fetch dependence
over heterogeneous surfaces in a physically consistent manner. To date this scheme is unconstrained
by observational data. Here, we validate the scheme and provide recommendations for its tuneable
parameters based on the joint ACSE and AO2016 data sets. E2016 pointed out that variation in the





morphological parameters $\beta$ and $c_e$ in L2012 could explain the variability of $C_{D10n}$ within concentration
bins. Reducing the values of $\beta$ and $c_e$ from those suggested by L2012 resulted in a better fit to their
data.
In the fetch dependent L2015 parameterization, increasing $\beta$ (sea-ice morphology exponent, Eq. 8 and
9) results in decreasing $C_{D10n}$, mostly at high ice concentrations, while increasing $c_e$ (the effective
resistance coefficient) increases $C_{D10n}$ at all concentrations. Here we have adjusted the L2015 values
of $\beta$ and $c_e$ to optimise the fit to our measurements. The revised values of the coefficients are given in
Table 1. For a consistent comparison, a similar tuning is applied to L2012.
Figure 8 shows $C_{D10n}$ plotted against $A_{Img}$ along with the tuned L2015 and L2012 schemes, both
anchored to the observed values of $C_{D10n}$ at $A_{Img} = 0$ and $A_{Img} > 0.8$ for the joint dataset.

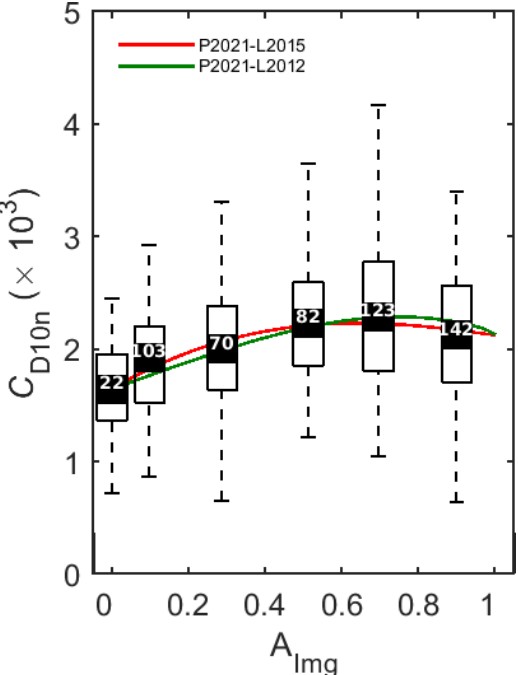


**Figure. 8** $C_{D10n}$ as function of local ice fraction ($A_{Img}$) for the joint ACSE and AO2016 (n = 542) dataset. Here,
the proposed parameterization (red line, P2021-L2015) is presented with L2015 anchored at the observed values
of $C_{D10nw}$ ($A_{Img} = 0$) and $C_{D10ni}$ ($A_{Img} > 0.8$) with coefficients $\beta$ and $c_e$ shown in Table 1. For comparison, the
L2012 schemes (green line, P2021-L2012) is also tuned with curve anchored at the same values of $C_{D10ni}$ and
$C_{D10nw}$ with $\beta$ and $c_e$ shown in Table 1.



Both L2012 and L2015 provide an excellent fit to the data, passing close the median observed values
at all ice fractions. The fitted curve for the joint dataset works equally well for the individual datasets
(Fig. S1, Supplementary material).
In the analysis above we have considered $C_{D10n}$ as a function of ice fraction – no distinction is made
between melt ponds and open water. However, there are uncertainties in the surface classification, in
particular for the determination of melt pond fraction. Thin ice and shallow melt ponds can appear
very similar in colour, and potentially be misclassified by the image processing algorithm. An
assessment of the sensitivity of the fitting of the L2015 to the presence and treatment of melt ponds
(see appendix B) shows that they have little impact.
Melt ponds are explicitly included in the L2012 and L2015 parameterizations in their more complex
levels of implementation, where the edges of melt ponds provide a source of form drag. Tsamados et
al. (2014) modelled the different contributions to the total drag using L2012 implemented within the
CICE sea ice model (Hunke and Lipscomb, 2010). They found that melt ponds made a negligible
contribution to the drag except over the oldest, thickest ice just north of the Canadian archipelago,
consistent with our observations.
Both L2012 and L2015 can be tuned to provide excellent fits to the observations (Fig 8). Even without
tuning to this data set, the differences between L2015, E2016A and E2016B are modest and all lie
within the interquartile range of the observed $C_{D10n}$ at all ice fractions (Fig 6a). The largest source of
uncertainty in the application of these schemes is the value of the drag coefficient at 100% ice fraction,
$C_{D10ni}$, which must be prescribed, and is strongly dependent on ice morphology. Table 2 lists values of
the neutral drag coefficient for very high ice fractions (0.8–1 from this study, and 0.9–1 or 1 in previous
studies) reported in the literature. The best estimates (mean or median values) vary by a factor of more
than 4. As discussed in previous studies, $C_{D10ni}$ depends on the sea-ice morphology and so prescribing
this as one value is a drastic simplification.







| $C_{D10ni}$ (×10⁻³) | | | N | Location/Morphology | Reference |
|---|---|---|---|---|---|
| Median or mean | Interquartile range or ±s | Full range | | | |
| 2.1 | 1.3–2.8 | 0.4–5.5 | 74 | Eastern Arctic (ACSE) | This study |
| 2.2 | 1.6–2.7 | 1.1–3.3 | 68 | Central Arctic (AO2016) | This study |
| 3.4 | 2.5–4.2 | 1.8–5.7 | 24 | Iceland Sea | Elvidge et al. (2021) |
| 2.6 | 2.4–3.9 | 1.9–4.0 | 8 | Barents Sea (broken floes) | Elvidge et al. (2016) |
| 0.9 | 0.4–2.1 | 0.1–3.8 | 32 | Fram Strait (large flat floes) | Elvidge et al. (2016) |
| 1.9 | 1.5–2.2 | | | Fram Strait (REFLEX I & II) | Lüpkes & Birnbaum (2005) |
| 1.5 | | | | Fram Strait (REFLEX) | Mai et al. (1995) |
| 3.8 | 2.5–5.1 | | | Very rough floes | Guest & Davidson (1987) |
| 1.5 | 1.2–1.9 | 1.2–1.9 | | Large flat floes | Table 2, Overland (1985) |
| | 1.7–2.6 | 1.7–3.7 | | Rough ice with ridges | Tables 3,6, Overland |
| | 2.2–2.7 | | | Marginal Seas, broken ice | Table 6, Overland |

**Table 2** Overview of neutral drag coefficients based on in situ eddy covariance measurements over 'complete' sea-ice cover ($C_{D10ni}$) from this and previous studies. The values are taken from the literature, so vary as to whether the mean, median or a range of values is shown. The definition of 'complete' sea ice covers a range of ice fractions (0.8–1, 0.9–1.0, and exactly 1) depending on the study. N is the number of data points in this bin where specified. The 2nd column provides the interquartile range or the range from -1s to +1s, where s is the standard deviation. Note Guest and Davidson (1987) uses the turbulence dissipation method. Overland (1985) compiles values from a variety of previous studies in various locations, as well as new data, so the values reproduced here are a compilation by morphology.

# 5    Conclusions

An extensive set of measurements of drag coefficients over sea ice, obtained during two research cruises within the Arctic Ocean, has been utilized to evaluate the dependence of drag on ice fraction. The final data set consists of 542 estimates of drag coefficients along with estimates of the local ice fraction obtained from high resolution imagery of the surface around the ship. The measurements cover a wide geographic area, summer melt and early autumn freeze up, and a range of surface conditions from thick multiyear ice, through melting ice with melt ponds, to newly formed thin and pancake ice, and near-surface stability conditions of $-2 < z/L < 1$, a much wider range than E2016. This wide range of conditions means the results should be broadly representative of much of the Arctic sea ice region.

The dependence of $C_{D10n}$ on ice fraction is evaluated in the context of recent, state-of-the-art parameterization schemes (Lüpkes et al., 2012; Lüpkes and Gryanik, 2015). The most recent of these (Lüpkes and Gryanik, 2015) attempts to account for the impact of short fetch over ice/water over



spatially highly heterogeneous surfaces. When tuned to the observations, the parameterizations provide
an excellent representation of $C_{D10n}$ as a function of ice fraction:
The main conclusions are:
• The data support the existence of a negatively skewed distribution of $C_{D10n}$ with ice
concentration, with a peak value for fractions of 0.6–0.8, consistent with the predicted
behaviour from Lüpkes et al. (2012) and observations of Elvidge et al. (2016).
• When tuned to our measurements, both L2012 and L2015 provide an excellent fit to the
observed variation of $C_{D10n}$ with ice fraction. The impact of small-scale surface heterogeneity
and the influence of fetch is likely to increase with increasing contrast in the skin temperatures
of the ice and water surfaces, and thus play a greater role in the winter.
• Melt ponds had no significant impact on the drag coefficient over the study area. The optimum
fit of the L2015 parameterization to the measurements had little sensitivity to the uncertainty
in partitioning of melt ponds to the ice or water fractions when estimating the local ice fraction;
and there was little sensitivity to the presence of melt ponds at all for the conditions observed.
• When evaluated against the AMSR2 retrieval of ice fraction, the behaviour of $C_{D10n}$ is not
consistent with in situ observations, for example, no peak is seen at intermediate ice fractions.
This is likely a result of several factors: a mismatch in spatial scale between the in situ flux
footprint (of order 100s of metres to 1 km) and the satellite footprint (6.5 km); potential spatial
offsets in location matching resulting from the low temporal resolution of the satellite data
(daily retrievals) combined with drifting of the ice; and the high scatter and varying mean bias
between the in situ and satellite estimates of ice fraction. The mean bias in particular displays
temporal/spatial coherence that suggests a dependence upon surface conditions. This finding
cautions against the use of comparatively low resolution remote-sensing products when
evaluating parameterizations.
Atmospheric stability may also play a role here, since it will affect how rapidly the atmospheric surface
layer adjusts to changes in surface properties. L2015 incorporates stability effects in the higher levels
of parameterization complexity, but not within the simplest complexity level used here. A much larger
data set, including the details of surface heterogeneity, would be required to evaluate the details of
both stability and fetch dependencies.
Sea ice and climate models are starting to incorporate components of form drag within their surface
exchange schemes for sea ice (e.g., Tsamados et al., 2014; Renfrew et al., 2019). But at present, most



do not use all the components of the more complex versions of schemes such as L2012 or L2015.
Instead, they tend to rely on the simplest versions where drag is only a function of ice fraction. In
operational forecast models, where only a prescribed ice concentration from a satellite retrieval may
be available, this seems appropriate, but within more complex coupled weather and climate prediction
models there is the potential for using output from the sea-ice model to adjust the drag coefficient
(E2016; Renfrew et al., 2019). The skill of the parameterization is strongly dependent on the accurate
representation of the drag at 100% ice fraction, $C_{D10ni}$, which varies significantly with ice morphology
(Lüpkes et al., 2012; Lüpkes and Gryanik, 2015; Elvidge et al., 2016, 2021). Tackling the
representation of $C_{D10ni}$ should be the next challenge in improving air-ice surface drag in weather and
climate models.



## Appendix A: Image processing and evaluation of local ice fraction

A total of ~500,000 images of the surface around the ship were obtained over the two cruises, so this required an automated approach to estimate the local ice fraction. Here we use the Open Source Sea-ice Processing (OSSP) algorithm of Wright and Polashenski (2018).

**(a) Pre-processing:**

Of the images available for each flux period, a subset of visibly good images was selected for further processing. The rejection of images was due to the presence of dense fog, moisture or ice on the camera lens, strong surface reflection of direct sunlight, or insufficient illumination. The selected subsets consist of 10 to 60 images in each flux period (e.g., Fig. A1(a)). These images are first corrected for lens distortion. The lens specific distortion coefficients and intrinsic parameters were determined using the Computer Vision System Toolbox of MATLAB. The corrected images (2048×1536 pixels) were then cropped to select a region within ~200m of the ship (2009×1111 pixels) – e.g., see Fig. A1(b).

**(b) Training and implementation of the algorithm:**

The success of any machine learning-based algorithm depends upon the quality of the training dataset. Since the ice conditions varied substantially throughout the campaigns, extensive training data was needed to cover the wide range of conditions. The initial training images selected were from the first and last image from each flux period. Additional images were added iteratively depending upon the performance of the algorithm on randomly selected images. After multiple trials, we settled on three different training datasets for (i) images with visibly large ice fraction, (ii) images with large open water fraction, (iii) images showing newly formed thin ice. Our approach was to generate a training data set that could be utilized equally on imagery from other campaigns, while keeping the number of discrete training datasets as small as possible. The training data, identifying ice, water, and melt-ponds was generated based on user classification of the training images via a Graphical User Interface (GUI), and thus depends upon the ability of the user to identify the surface features correctly.

**(c) Post-processing:**

The OSSP algorithm produces an indexed image having pixel-wise information about surface features (open water, melt ponds, ice) for each input image (e.g., Fig. A1(c)). Since the images were necessarily taken at an oblique angle, the indexed images need to be orthorectified to derive the correct fractions of ice, melt pond and water. Orthorectification of imagery is a process by which pixel elements of an oblique image are restored to their true vertical perspective position. The angular separation of each

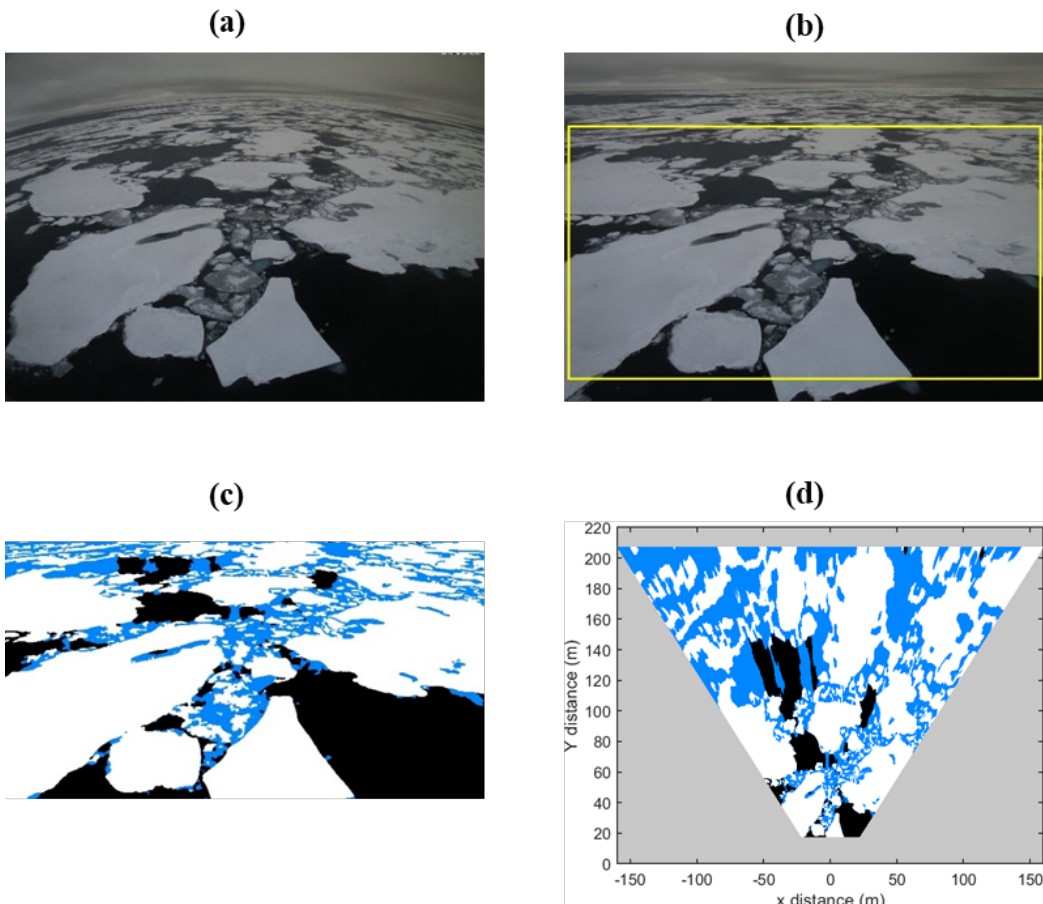

546

547

**Figure A1.** An example of the image processing workflow. Panel (a) is an example raw image; (b) shows the image corrected for the lens distortion, where the region of focus is shown by the yellow rectangle; (c) shows the image after processing by the OSSP algorithm where the masking colours – white, blue and black – represent ice, melt pond/submerged ice and open water areas, respectively; (d) shows the orthorectified image showing the true distance of each surface feature away from the camera.


pixel (after correction for lens distortion) was determined from a lab calibration of the cameras. The
angle from the horizon (the horizontal) in the images, and the height of the camera above the surface
then allows the location of each pixel on the surface to be calculated. The masked images were
interpolated onto a regular $x$-$y$ grid after orthorectification and area fractions of ice/melt ponds were
estimated as a fraction of the total number of pixels for each category (e.g., Fig. A1(d)). The average



fractions of ice/water for a flux period are then calculated by taking an average over all the images in
that period. Only flux periods having more than 30 available images are included in the analysis.

## Appendix B: Sensitivity to melt ponds

Here we investigate the sensitivity of our tuning of L2015 to the melt pond fraction. We reclassify
50% and 100% of melt ponds as ice instead of water (Fig. B1) and the L2015 function is re-fitted to
the revised ice fractions and compared with our original fit.

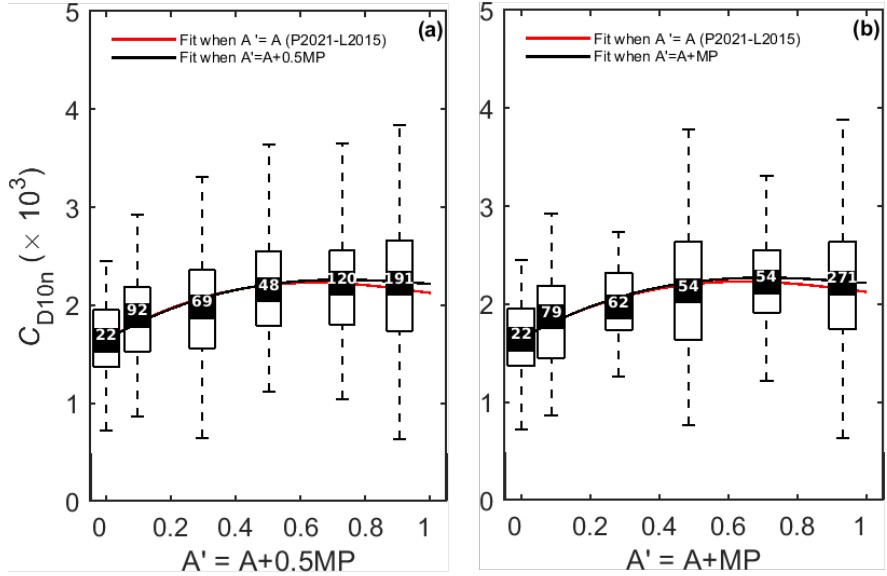


**Figure B1.** Parameterization sensitivity to melt pond fraction (MP). Panel (a) re-classifies half of melt ponds to
sea ice (A'=A+0.5*MP); while (b) re-classifies all of melt points to sea ice (A'=A+MP), where A is sea-ice
fraction. The curves show the L2015 parameterization, tuned to the original ice fraction observations (red;
P2021-L2015) and tuned to the adjusted ice fraction observations (black).


The reclassification of melt ponds as ice has the effect of moving some drag estimates into higher ice
fraction bins, slightly increasing the median value of $C_{D10n}$ at high ice fractions. The refitted L2015
functions reflect this slightly higher drag at high ice fraction but are essentially unchanged for $A < 0.5$.
Note even when $A > 0.5$ the change in the L2015 functions is very small compared with the variation
in $C_{D10n}$ within each ice concentration bin. We further investigate the sensitivity of the parameterization
to the presence of melt ponds by a simple sub-setting of the data by melt pond fraction. In all cases the
melt pond fraction is $< 0.6$. Figure B2 shows $C_{D10n}$ with ice fraction for cases where the melt pond





fraction is < 0.3 (Fig. B2a) and < 0.1 (Fig. B2b). The L2015 function is fit to these subsets of data and
compared with that to the full data set. The revised fits differ negligibly from that to the full data set,
suggesting that $C_{D10n}$ is not strongly dependent on the extent of melt ponds. In short, the sensitivity of
the parameterization to the treatment of melt ponds is negligible.

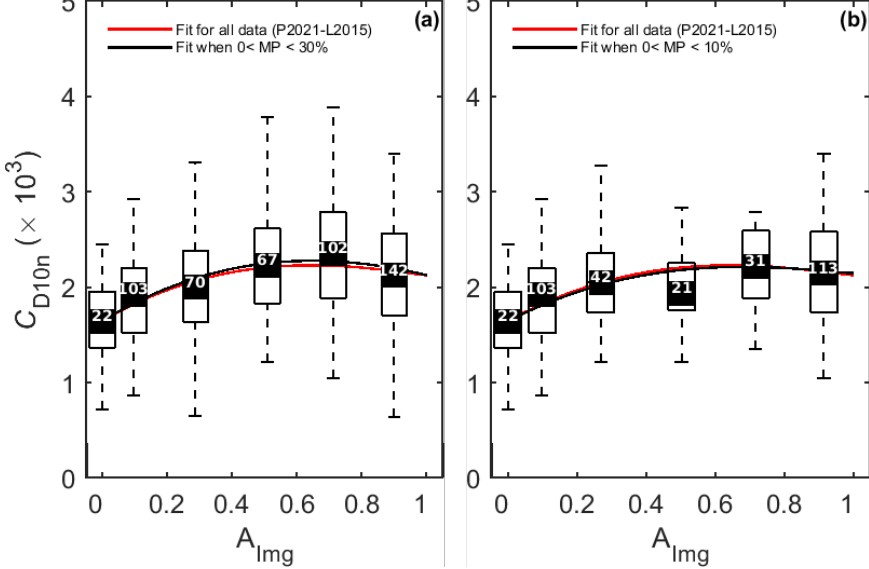


**Figure B2.** (a) MP< 30% (b) MP<10%. In each panel, the red curve is the fitting curve obtained for the joint
ACSE and AO2016 data same as shown in Figure 8 (P2021-L2015) and the black lines are the fitting curve
obtained for the data shown in the 'respective' panels.




*Data Availability*. The UK ACSE cruise data is publicly available from the Centre for Environmental
Data Analysis (CEDA) archives (http://archive.ceda.ac.uk). Data from all participants in both cruises
is publicly available from the Bolin Centre for Climate Research data archive (https://bolin.su.se/data).
*Author Contributions*. All authors contributed to the design of the study. PS analysed the data sets
and wrote the manuscript with contributions from all co-authors. IMB, JP, and DJS collected the data
during ACSE; IMB and JP collected the data during AO2016. JP processed the flux data. DJS and PS
processed the surface imagery.
*Competing interests*. The authors declare no competing interests.
*Acknowledgements*. We would like to thank the captains and crews of the icebreaker *Oden*, along with
the technical and logistical support staff of the Swedish Polar Research Secretariat, for their assistance
throughout the ACSE and AO2016 cruises. We thank Michael Tjernström, Ola Persson, Matthew
Shupe, Barbara Brooks, Joseph Sedlar, and Georgia Sotiropoulou for their contributions to the ACSE
measurement campaign.
*Financial support*. This work was funded by the UK Natural Environment Research Council (NERC)
grant numbers NE/S000453/1 and NE/S000690/1. MJY was also supported by NERC grant numbers
NE/N018095/1 and NE/V013254/1. The contribution of IMB, JP, and DJS to the ACSE cruise was
funded by NERC grant number NE/K011820/1. Participation in the AO2016 cruise was supported by
the Swedish Polar Research Secretariat.




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
