# Peer review of "Ship-based estimates of momentum transfer coefficient over"

_Atmospheric Chemistry and Physics, 2021_

## Referee Comment (RC1)

**Review of**

**Ship-based estimates of momentum transfer coefficient over sea ice and recommendation for its parameterization by Srivastava et al.**

This paper describes measurements of the near-surface momentum fluxes obtained from two ship cruises to the Arctic in summers 2014 and 2016. Momentum transfer coefficients are derived and results are compared with existing parameterizations accounting for the form drag of floe edges. It is shown that the drag coefficients peak at sea ice concentration of 0.6-0.8, which was also postulated by the parameterizations. After some tuning, two parameterizations show an impressing agreement with the measurements. It requires enormous logistic efforts to gain measurements like those presented in the manuscript and so they are unique in the literature.

The paper is very well written and follows a clear logic. Results are very useful for modellers. So, I have only a small number of minor revisions and can suggest the publication of the text with only little modifications.

**Minor Revisions**

- 1. Equation 1: It is a simplification because the small term  $+\psi_m(z_0/L)$  in brackets is neglected. Please tell also that  $\zeta = z/L$  where L is the Obukhov length.
- 2. Equation 12: I suggest writing below the equation something that  $U_{10n} = U_{10}f_m$ where  $f_m$  is the stability correction that can be derived from the Businger-Dyer stability correction.
- 3. Section 3.4: It is explained here and later that two sets of sea ice fractions were used. The size of the representative area for the sea ice fraction, especially from the onboard imagery, is not completely clear for me. Ideally, the parameterizations would require a region of perhaps 5-10 km diameter. But this value is rather vague and depends probably also on the situation. E.g. it might depend on the floe sizes and homogeneity. Could you include a few sentences about this? What is your opinion, based on the data and footprint, about the ideal size of the region?

---

## Author Response (AR1)

**Reply to Reviewer 1**

*Comment 1: This paper describes measurements of the near-surface momentum fluxes obtained from two ship cruises to the Arctic in summers 2014 and 2016. Momentum transfer coefficients are derived, and results are compared with existing parameterizations accounting for the form drag of floe edges. It is shown that the drag coefficients peak at sea ice concentration of 0.6-0.8, which was also postulated by the parameterizations. After some tuning, two parameterizations show an impressing agreement with the measurements. It requires enormous logistic efforts to gain measurements like those presented in the manuscript and so they are unique in the literature. The paper is very well written and follows a clear logic. Results are very useful for modellers. So, I have only a small number of minor revisions and can suggest the publication of the text with only little modifications.*

Reply: We thank the reviewer for encouraging remarks on the manuscript. We have carefully gone through the comments/suggestions and revised the manuscript accordingly.

*Comment 2: Equation 1: It is a simplification because the small term $+\psi_m(z_0/L)$ in brackets is neglected. Please tell also that $\zeta = z/L$ where $L$ is the Obukhov length.*

Reply: Thank you for pointing out this. $\psi_m(z_0/L)$ is not included in the derivation of $C_D$. The equation is, however, now modified in the revised manuscript to be shown in full, and the fact that $\psi_m(z_0/L)$ is neglected is noted.:

$$C_D = \kappa^2 [ln(z/z_0) - \psi_m(z/L) + \psi_m(z_0/L)]^{-2}$$

*Comment 3: Equation 12: I suggest writing below the equation something that $U_{10n} = U_{10}f_m$ where $f_m$ is the stability correction that can be derived from the Businger-Dyer stability correction.*

Reply: The sentence is modified as:

'where $u_*$ is the measured friction velocity, and $U_{10n}$ is the 10 m equivalent neutral wind speed corresponding to 10 m wind speed $U_{10}$, determined using Monin-Obukhov similarity theory and Businger-Dyer stability correction function $f_m$ (Businger et al., 1971) as $U_{10n} = U_{10}f_m$.'

*Comment 4: Section 3.4: It is explained here and later that two sets of sea ice fractions were used. The size of the representative area for the sea ice fraction, especially from the onboard imagery, is not completely clear for me. Ideally, the parameterizations would require a region of perhaps 5-10 km diameter. But this value is rather vague and depends probably also on the situation. E.g. it might depend on the floe sizes and homogeneity. Could you include a few sentences about this? What is your opinion, based on the data and footprint, about the ideal size of the region?*

Reply: An individual image from onboard imagery was cropped to select a region within ~200m of the ship. Each cropped image has an area of approximately 34,225 $m^2$. The ship was mostly moving and capturing the surface features for a 30-min time duration. With typical ship speeds

through the ice in the range of 2-5 m/s, depending on ice thickness, a half-hourly flux period would be representative of a region of ~3.5 to 9 km along track. The total area imaged for each 30-minute flux estimate depends on the number of images passing quality control, but reaches maxima of up to approximately 2.05 km$^2$ for ACSE (60 independent, non-overlapping images at 1 image per minute from both cameras) and about 6.7 km$^2$ for AO2016 (up to 240 images, overlapping by 75 m along track at a ship speed of 5 m s$^{-1}$). We have to assume here that the ice fraction determined along the ship track is representative of that within the flux footprint. The fact that the behavior of the drag coefficient as a function of local ice fraction behaves more consistently than as a function of satellite-derived ice fraction (with a much larger averaging area) suggests that this assumption is reasonable.

Brief details of the image area have been added to section 3.4, and a more detailed explanation added to Appendix 1.

**Reply to Reviewer 2**

*Comment 1: This paper presents ship-based measurements of near-surface momentum fluxes obtained from two field campaigns namely Arctic Clouds in Summer Experiment (ACSE; July-October 2014) and Arctic Ocean 2016 experiment (AO2016; August-September 2016). Authors have presented over 500 new estimates of surface drag and local sea-ice concentration measurements derived from onboard imagery. The datasets presented here are much larger than those documented in the literature to date and are supposed to be representative of much of the Arctic sea-ice region. This unique dataset is utilized to investigate the relationship between surface drag and sea-ice concentration within the framework suggested by Lupkes et al. (2012), Elvidge et al. (2016), and Lupkes and Gryanik (2015). It is shown that with minor tuning two parameterizations are in well- agreement to the measurements.*

*Firstly, I would like to emphasize that these types of observations over the Arctic are rare, and it requires huge efforts to collect, process, and analyze measurements like this. Apart from that, processing the over ~ 500, 000 sea-ice images to derive the local sea-ice concentration for each flux period requires enormous effort.*

*The paper is very interesting and well written, and the authors have brought out the novelty of the study in a logical manner. I have only a few minor comments and suggest publication after minor modification.*

*Reply*: We thank the reviewer for encouraging remarks on the manuscript. We have carefully gone through the comments/suggestions and revised the manuscript accordingly.

*Comment 2: Line 74-75: References should be in order.*

Reply: Corrected.

*Comment 3: Is the second dataset Arctic-Ocean 2016 (AO2016) is utilized for the first time for scientific publication? If not, I would suggest adding a reference.*

Reply:  The AO2016 data is used here for the first time.

*Comment 4: Line 179: Field **m**easurements*

Reply: Corrected.

*Comment 5: Line 190 and other places: eddy-covariance*

Reply: Corrected.

*(5) Line 197: I suggest computational fluid dynamics (CFD) model*

Reply: Corrected.

*Comment 6: Line 211: 7$^{th}$*

Reply: Corrected.

*Comment 7: Line 223: 10-m*

Reply: Corrected.

*Comment 8: Line 258: Satellite-based*

Reply: Corrected.

*Comment 9:  Line 268: References should be in order.*

Reply: Needful is done.

*Comment 10:  Line 289-291: I think this needs rephrasing.*

Reply: Lines 289-291 read: "Figures 2 and 3 show the meteorological and surface conditions during ACSE and AO2016. The first half of ACSE was dominated by relatively low winds, and surface temperatures close to 0°C when in the ice; much warmer temperatures are associated with open coastal waters. The second half of the…" – we can't see what the issue might be here. Perhaps the wrong line numbers are cited?

*Comment 11: Line:382: 'at low an ice concentration'*

Reply: The review has misread (and misquoted) the text, which actually reads "…at too low an ice concentration" which is grammatically correct and conveys the intended meaning

Comment 12: *Line 424: Merely presenting a Figure in the supplementary material doesn't look well. I suggest adding a few lines to explain it.*

Reply: The following lines have been added to describe the figure:

'*Figure S1 shows the best fits of the L2015 parameterization to the individual data sets for ACSE and AO2016 (black lines) along with the best fit to the joint data set, as derived in the main paper (P2021-L2015, red line). L2015 provides an excellent fit to both individual data sets, passing close the median observed values at all ice fractions, and differs little in either case from the fit to the joint data sets.*'

Comment 13: *Table2: This table looks quite interesting and informative. Please correct it-Overland (1985).*

Reply: The reference of Overland (1985) has been added at both places in the Table.

Comment 14: *Beyond the scope of this paper, I hope to see the validity of the stability-dependent form of the L2015 scheme in future studies. Further, high-quality measurements like this could also be utilized for the analysis of scalar transfer.*

Reply: We are presently working on the parameterization of scalar transfer coefficients using both the ACSE and AO16 datasets. The initial results for the heat transfer coefficient are quite promising and found largely to support the conclusions of Elvidge et al. (2021). The analysis of the stability-dependent form of L2015 is a challenging problem within the framework and limitations of our datasets. The scheme requires separate temperature measurements for ice and water fractions to incorporate the joint effects of stability due to both surfaces over MIZ. We hope to simplify things further and look into the validity of the stability-dependent form of the L2015 scheme in our future studies.